# A Genome-Wide Identification and Expression Pattern of *LMCO* Gene Family from Turnip (*Brassica rapa* L.) under Various Abiotic Stresses

**DOI:** 10.3390/plants12091904

**Published:** 2023-05-07

**Authors:** Waqar Khan, Ahmed M. El-Shehawi, Fayaz Ali, Murad Ali, Mohammed Alqurashi, Mohammed M. Althaqafi, Siraj B. Alharthi

**Affiliations:** 1State Key Laboratory of Crop Genetics and Germplasm Enhancement, Nanjing Agricultural University, Nanjing 210095, China; waqar.khan399@gmail.com; 2Department of Biotechnology, College of Science, Taif University, P.O. Box 11099, Taif 21944, Saudi Arabiamm.mohammad@tu.edu.sa (M.M.A.); 3Department of Botany, Shaheed Benazir Bhutto University, Sheringal, Dir Upper P.O. Box 18050, Pakistan; 4Department of Botany, Hazara University Mansehra, Mansehra P.O. Box 21300, Pakistan; 5Department of Biological Sciences, King Abdulaziz University, P.O. Box 16834, Jeddah 21474, Saudi Arabia

**Keywords:** identification, phylogeny, abiotic stress, gene expression pattern, *B. rapa*, LMCO gene family

## Abstract

Laccase-like multi-copper oxidases (LMCOs) are a group of enzymes involved in the oxidation of numerous substrates. Recently, these enzymes have become extremely popular due to their practical applications in various fields of biology. LMCOs generally oxidize various substrates by linking four-electron reduction of the final acceptor, molecular oxygen (O_2_), to water. Multi-copper oxidases related to laccase are extensively distributed as multi-gene families in the genome sequences of higher plants. The current study thoroughly investigated the LMCO gene family (Br-Lac) and its expression pattern under various abiotic stresses in *B. rapa* L. A total of 18 Br-Lac gene family members located on five different chromosomes were identified. Phylogenetic analysis classified the documented Br-Lac genes into seven groups: Group-I (four genes), Group-II (nine genes), Group-III (eight genes), Group-IV (four genes), Group-V (six genes), and Group-VI and Group-VII (one gene each). The key features of gene structure and responsive motifs shared the utmost resemblance within the same groups. Additionally, a divergence study also assessed the evolutionary features of Br-Lac genes. The anticipated period of divergence ranged from 12.365 to 39.250 MYA (million years ago). We also identified the pivotal role of the 18 documented members of the LMCO (Br-lac) gene family using quantitative real-time qRT-PCR. Br-Lac-6, Br-Lac-7, Br-Lac-8, Br-Lac-16, Br-Lac-17, and Br-Lac-22 responded positively to abiotic stresses (i.e., drought, heat, and salinity). These findings set the stage for the functional diversity of the LMCO genes in *B. rapa*.

## 1. Introduction

Multi-copper oxidases (MCOs) are a group of enzymes from the kingdom Fungi that contain laccases and peroxidases [1,2], metal oxidases from bacteria [3], and ascorbate oxidases from the kingdom Plantae [4]. LMCOs, commonly known as “laccases”, are mainly involved in the oxidation of industrial substrates as well as in developmental processes in plants. Some of the prominent developmental processes include polymerization of lignin and flavonoids and anthocyanin degradation. These are very helpful in the browning of the pericarp during postharvest in litchi fruit [5]. Lignin digestion in fungi [6], metabolic activities (e.g., pigment production), and several other activities, such as pathogenic virulence in bacteria and plant species, are other key features of these enzymes [7,8]. Metal transportation and homeostasis in bacterial cells [9,10], healing of wounds, and lignin production in higher plants are also associated with laccases [11,12].

The genomes (transcriptomes) of different members of the kingdom Plantae (woody and herbaceous) provide valuable information for identifying oxidative enzymes from the laccase family. Genomes of different plant species, including *Prunus avium* L., *Trametes trogii*, *Arabidopsis thaliana*, *Liriodendron tulipifera*, and *Setosphaeria turcica*, etc., have been extensively studied for the identification of these LMCO genes. However, there is a dearth of studies about the LMCO genes in members of the family Brassicaceae. Several members of this family are being utilized as vegetables, oilseed, and feed crops across the globe, and they play an important role in uplifting living conditions of concern. According to “whole genome triplication” (WGT), *Brassica* and *Arabidopsis* diverged from a common ancestor approximately 20 MYA. Similarly, *B. rapa* and *B. oleracea* came into existence from a common ancestor approximately 3.75 MYA [13,14]. The genomic materials in *Brassica* species have increased through whole genome triplication (WGT) and tandem duplication (TD), which make it an excellent model [14]. *B. rapa* L. is an economically important member of Brassicaceae, formerly known as Cruciferae. This plant species is an important source of vegetables that are rich in carotenoids (lutein, zeaxanthin, etc.), β-carotene, glucosinolates, iron, and calcium contents [15].

Five subgroups of the LMCO gene family have been reported through functional analysis in *A. thaliana*. At-Lac-12, At-Lac-13, and At-Lac-22 have been reported to enhance drought tolerance in *A. thaliana* [16]. Drought and salt stress tolerance in *Oryza sativa* were enhanced by overexpressing Os-LAC-6 and Os-LAC-8 [17]. LAC-12, LAC-13, and LAC-22 promote drought resistance, while LAC-6 and LAC-8 promote salinity resistance in *A. thaliana* [17]. Previous studies have shown that the LMCO gene family enhances tolerances under different stress conditions in various plants, but no such studies exist for the genome of *B. rapa* L. [14,18]. In the current study, we attempted to analyze the phylogeny, domain, motif, gene structure, chromosome placement, gene duplication, and gene expression of the LMCO (Br-Lac) gene family in *B. rapa*. This study’s findings will help lay the groundwork for future research on its biological function.

## 2. Results

### 2.1. Identification and Classification Pattern of the LMCO Gene Family in B. rapa

We identified 18 Br-Lac (LMCO) genes in *B. rapa*, which were further confirmed from the SMART and NCBI websites for domain validation. The names for the identified genes were given in sequence (e.g., from Br-Lac-1 to Br-Lac-17 in a row and Br-Lac-22). The nomenclature was purely based on information from *A. thaliana* due to sequence similarity. The phylogenetic tree divided the 18-member LMCO gene family into seven groups: Group-I (Br-Lac-2 and Br-Lac-17), Group-II (Br-Lac-4, Br-Lac-10, Br-Lac-11, and Br-Lac-22), Group-III (Br-Lac-3, Br-Lac-5, Br-Lac-12, and Br-Lac-13), Group-IV (Br-Lac-14 and Br-Lac-15), Group-V (Br-Lac-7, Br-Lac-8, and Br-Lac-9), Group-VI (Br-Lac-1), and Group-VII (Br-Lac-6) (Figure 1). A Appendix A provides a detailed list of the current classifications.

### 2.2. Synteny Analysis of Br-Lac Genes

The 18 documented putative LMCO genes were mapped on five different chromosomes: chromosome A02 consists of Br-Lac-10–Br-Lac-12 and Br-Lac-15. Chromosome A03 consists of Br-Lac-6, Br-Lac-9, and Br-Lac-17. Chromosome A05 consists of Br-Lac-2–Br-Lac-4, Br-Lac-7, and Br-Lac-22, whereas chromosome A01 contains only one of the reported genes (i.e., Br-Lac-1). Chromosome A10 consists of Br-Lac-13, Br-Lac-14, and Br-Lac-16 (Figure 2).

### 2.3. Classification of A. thaliana and B. rapa LMCO Genes

A maximum likelihood (ML) phylogenetic tree was constructed for the 35 reported LMCO genes (i.e., 17 from *A. thaliana* (At-Lac) and 18 from *B. rapa* (Br-Lac)). The ML phylogenetic tree divided all the documented 35 LMCO genes into seven groups (Appendix A).

The protein lengths of the Br-Lac genes from *B. rapa* ranged from 558 to 582 amino acids (aa). The molecular weights of the resultant proteins were between 61.22 and 65.75 kDa, while pIs varied from 6.07 to 9.73. These proteins possess a negative grand average of hydropathicity (GRAVY), indicating hydrophilic behavior. Understanding plant functions requires knowledge of subcellular localization, and our results showed that most of the proteins were found in the chloroplast and vacuoles (Table 1).

Additionally, the exon–intron structures of the 35 LMCO genes were positioned according to their phylogenetic relationships. LMCO genes of the same group showed more similarities with the LMCO protein sequences. The exon and intron numbers differed between the LMCO genes of *A. thaliana* and those of *B. rapa*. The number of exons in the At-Lac and Br-Lac genes was five to seven in *A. thaliana* and *B. rapa* (Figure 3).

### 2.4. Conserved Motif Distribution

Fifteen conserved motifs were identified in *A. thaliana* and *B. rapa* LMCO proteins using the MEME tool and MEME suite web server to acquire the logos (Figure 4; Appendix A). Motifs 1–8, 10, 12, and 15 were conserved in all LMCO genes of *A. thaliana* and *B. rapa*. Motif 9 was conserved, except in Br-Lac-1, while Motif 11 was missing in At-Lac-14 and Br-Lac-14. Motif 13 was conserved in all LMCO genes of *A. thaliana* and *B. rapa*, except At-Lac-6 and Br-Lac-6. Motif 14 was found in At-Lac-1–At-Lac-17, except for At-Lac-15, and in Br-Lac-1–Br-Lac-17, except for Br-Lac-15 and Br-Lac-22. Motif 9 was found in At-Lac-1, At-Lac-2, At-Lac-3, At-Lac-4, At-Lac-5, At-Lac-6, At-Lac-7, At-Lac-8, At-Lac-9, At-Lac-10, At-Lac-11, At-Lac-12, At-Lac-13, At-Lac-14, At-Lac-15, At-Lac-16, and At-Lac-17 from Arabidopsis and Br-Lac-2, Br-Lac-3, Br-Lac-4, Br-Lac-5, Br-Lac-6, Br-Lac-7, Br-Lac-8, Br-Lac-9, Br-Lac-10, Br-Lac-11, Br-Lac-12, Br-Lac-13, Br-Lac-14, Br-Lac-15, Br-Lac-16, Br-Lac-17, and Br-Lac-22, while motif 9 was missing in Br-Lac-1.

### 2.5. Calculating the Non-Synonymous (Ka) and Synonymous (Ks) Substitution Rates

As part of the divergence analysis, the non-synonymous substitution per non-synonymous site (Ka) and synonymous substitution per synonymous site (Ks) for each pair of paralogous Br-Lac genes were calculated according to the phylogenetic tree server (Appendix A). This was carried out to understand the degree of evolutionary discretion among Br-Lac genes. A Ka/Ks value lower than 1 demonstrated the presence of purifying selection pressure during evolution. Each pair of Br-Lac genes went through a period of divergence approximately 12.365 to 39.250 million years ago (Table 2).

### 2.6. Protein Structure Analysis of Br-Lac Genes

To obtain a more in-depth comprehension of the framework of the Br-Lac genes, the secondary and tertiary structures of the Br-Lac proteins were investigated (Table 3, Figure 5). Both folding and coiling helped in understanding the secondary structure of the Br-Lac proteins. The secondary structure of the Br-Lac proteins was made up of four basic components: the helix (H%), the turn (T%), the extended chain (E%), and the random coil (RC%). Following the helix (H%), which varied from 12.2% (Br-Lac-9) to 21.11% (Br-Lac-9), the random coil (RC%) in the secondary structure of Br-Lac proteins had the greatest value, ranging from 36.67% (Br-Lac-10) to 45.7% (Br-Lac9). The extended chain (E%) ranged from 28.77% (Br-Lac-3) to 32.99% (Br-Lac-1). The secondary structure of Br-Lac proteins was also supported by the tertiary structures of the proteins, as shown using ExPASy (https://swissmodel.expasy.org/) (accessed on 7 January 2023).

### 2.7. Expression Patterns of LMCO Genes under Abiotic Stress

We exposed *B. rapa* seedlings to various abiotic stresses (drought, salinity, and heat) to expose the expression patterns of its 18 documented LMCO genes. QRT-PCR was performed on the 18 Br-Lac genes at different time intervals after various abiotic treatments, and the expression levels were calculated (Figure 5). The expression patterns of the 18 Br-Lac genes showed transcriptional changes under abiotic stress. It was concluded that members of the LMCO gene family show a response to multiple stresses. Five genes (Br-Lac-1, Br-Lac-2, Br-Lac-6, Br-Lac-7, and Br-Lac-8) of the 18 Br-Lac genes were noted to have low expression levels in response to drought stress. At different time intervals, the expression levels of these five genes were down-regulated, while four genes (Br-Lac-4, Br-Lac-5, Br-Lac-16, and Br-Lac-17) were up-regulated. The expression level of Br-Lac-5 was increased at 12 h and 24 h but suddenly reversed at 48 h and 72 h. After drought treatment, the up-regulated genes (Br-Lac-4, Br-Lac-5, Br-Lac-16, and Br-Lac-17) may be related to drought tolerance (Figure 6).

Different expression levels of the 18 Br-Lac genes were observed under salinity stress conditions; seven genes (Br-Lac-1, Br-Lac-3, Br-Lac-9, Br-Lac-12, Br-Lac-14, Br-Lac-15, and Br-Lac-16) under salinity stress were up-regulated. The Br-Lac-12 gene showed high regulation at 12 h, while Br-Lac-14 showed up-regulation at 24 h (Figure 7). Three genes, Br-Lac-6, Br-Lac-7, and Br-Lac-17, were noted to have down-regulation due to salinity stress at different time intervals. The up-regulation of different genes (Br-Lac-1, Br-Lac-3, Br-Lac-9, Br-Lac-12, Br-Lac-14, Br-Lac-15, and Br-Lac-16) under salinity stress at varied time intervals suggests that some Br-Lac genes may be induced under serious salinity stress. Similarly, seven Br-Lac genes were up-regulated (Br-Lac-4, Br-Lac-8, Br-Lac-9, Br-Lac-10, Br-Lac-13, Br-Lac-14, and Br-Lac-16) upon exposure to heat. The different expression levels can be seen in Figure 8, at different time intervals, under heat stress treatment. At the same time, only three genes were down-regulated under heat stress conditions (Br-Lac-2, Br-Lac-6, and Br-Lac-7). Noteworthy, our findings suggest that the expression levels of Br-Lac-4, Br-Lac-9, and Br-Lac-16 were induced by drought, heat, and salinity stresses, suggesting that these genes might be important for resistance to abiotic stresses. A list of the primers is given in Appendix A.

## 3. Discussion

Determining the physiological functions of LMCOs is comparatively difficult due to their wide distribution and complex nature across various plant species. These factors, in combination with several others, make it very challenging to identify LMCO genes and their functions. Differentiating the xylem of *Liriodendron tulipifera*, *Pinus taeda*, *Zinnia elegans*, and *Acer pseudoplatanus* was used to isolate these enzymes [19,20,21,22]. Lignin monomers were oxidized in vitro by these enzymes, obtained from *Pseudoplatanus* cells, when maintained in suspension culture [23]. Prototype LMCOs (laccase) from *Rhus vernicifera* were found to be good in healing wounds caused either by herbivores or in response to pathogens [20,24]. The trunks of *Japanese lacquer* and *Rhus vernicifera* were reported to exhibit an oxidative polymerization reaction between laccase and the alkyl catechols in latex sap, causing a strong protective seal to form over the injury. This happens because laccase is a component of latex sap [20,25]. Furthermore, biochemical evidence also supports the idea that plant LMCOs play a role in iron metabolism [21].

A genome-wide gene family study is the initial step in understanding gene structure, function, and evolution [26]. Additionally, sequence-based searching and phylogenetic characterization are the most efficient techniques for identifying laccase genomes [27,28,29,30,31]. We carried out a thorough search for LMCO genes across the *B. rapa* genome and found a total of 18 genes. Based on domain organization and evolutionary analyses, these genes were further classified into seven subgroups (Figure 1). According to a synteny study, *B. rapa* and *A. thaliana* LMCO genes have significant similarities (Figure 3 and Figure 4). These results are remarkably consistent with those of earlier studies [20,32].

Isoelectric focusing points (pIs) have been used as the primary classification method for plant LMCOs, with the implicit belief that pI may be related to substrate kinetics and enzyme activity [6,20]. A search for the conserved domain, motif, and structure was carried out to better understand the shared characteristics and biological roles of LMCO genes (Figure 3 and Figure 4). The persistence of ancient and more modern gene duplication events showed that both purifying and diversifying selection lead to the emergence of new gene functions over time [32,33,34,35]. The current work identified and analyzed 18 LMCO genes in *B. rapa* and 17 LMCO genes in *A. thaliana*.

Biotic and abiotic stresses define the patterns of both gene expression and function [36]. QRT-PCR has been used in several research studies to investigate transcript levels, such as the expression profile of At-Lac-12 and At-Lac-14 in response to abiotic stresses. *A. thaliana* LMCO proteins were successfully involved in the process of drought tolerance [37]. To obtain further insights into the regulation of Br-Lac in response to abiotic stress, gene expressions in *B. rapa* were analyzed using real-time RT-PCR, as shown in Figure 6, Figure 7 and Figure 8. Four genes (Br-Lac-4, Br-Lac-5, Br-Lac-16, and Br-Lac-17) were up-regulated at different time intervals. For gene Br-Lac-5, the expression level was increased at 12 h and 24 h, and sudden down-expression was noted at 48 h and 72 h. Seven genes (Br-Lac-1, Br-Lac-3, Br-Lac-9, Br-Lac-12, Br-Lac-14, Br-Lac-15, and Br-Lac-16) under salinity stress were up-regulated. Under heat stress conditions (at different time intervals), seven Br-Lac genes were up-regulated (Br-Lac-4, Br-Lac-8, Br-Lac-9, Br-Lac-10, Br-Lac-13, Br-Lac-14, and Br-Lac-16). Zhang et al. [38] reported that expressions of the At-Lac-12 and At-Lac-14 genes were very limited in heat and drought stress conditions. In another study, At-Lac-17 and At-Lac-22 overexpression promoted drought resistance in *A. thaliana*, whereas the expression levels of At-Lac-9, At-Lac-11, and At-Lac-15 were inhibited by drought and salinity stress.

There has only been relatively little research on the LMCO gene family in plants. The LMCO gene family’s intron–exon configurations may cause functional variability. Such findings could be attributable to a homology structure separate from the domain sequences [21,33]. This study established a baseline for understanding the molecular function and response to abiotic stresses of Br-Lac genes and recommends future research for examining these genes concerning various biological functions.

## 4. Materials and Methods

### 4.1. Identification of LMCO Genes in A. thaliana and B. rapa L.

The protein sequences of the LMCO genes in *A. thaliana*, i.e., At-Lac-1 (At1g18140), At-Lac-2 (At2g29130), At-Lac-3 (At2g30210), At-Lac-4 (At2g38080), At-Lac-5 (At2g40370), At-Lac-6 (At2g46570), At-Lac-7 (At3g09220), At-Lac-8 (At5g01040), At-Lac-9 (At5g01050), At-Lac-10 (At5g01190), At-Lac-11 (At5g03260), At-Lac-12 (At5g05390), At-Lac-13 (At5g07130), At-Lac-14 (At5g09360), At-Lac-15 (At5g48100), At-Lac-16 (At5g58910), and At-Lac-17 (At5g60020), were searched and downloaded using TAIR10 (http://www.arabidopsis.org/) (accessed on 20 December 2022). These sequences were then used as queries to implement BLASTP searches in the *B. rapa* genome to find LMCO gene sequences. Phytozome was used to search and download Brassica genome sequences (http://phytozome.jgi.doe.gov/pz/portal.html#) (accessed on 22 December 2022). A Hidden Markov Model (HMM) profile for laccase (IPR017761) was obtained from the Pfam database (http://pfam.xfam.org/) (accessed on 29 April 2023) using HMMER3 software to search for LMCO homologous genes [39]. After removing all the redundant sequences, the SMART (http://smart.embl-heidelberg.de/) (accessed on 25 December 2022) and Pfam databases were used to examine the required sequences [40]. The physicochemical properties of the full-length proteins were assessed using the ProtParam tool (http://web.expasy.org/protparam/) (accessed on 28 December 2022) [41].

### 4.2. Phylogenetic Analysis

A phylogenetic tree for the full-length LMCO protein sequences of *A. thaliana* and *B. rapa* was constructed using MEGA software version 7.0. [42]. The ClustalW program was used to align the full-length LMCO protein sequences. The pairwise deletion option was selected, and the Poisson model with a 1000 bootstrap sample was used by applying the maximum likelihood method to make an LMCO tree.

### 4.3. Localization of the Chromosomes and Synteny Analysis

We examined genome data for the location of LMCO genes on the *B. rapa* chromosome. Syntenic relationships in the *Brassica* genomes were constructed using a method similar to the Plant Genome Duplication Database (PGDD; http://chibba.agtec.uga.edu/duplication/) (accessed on 28 December 2022) [43]. The syntenic link between the genes of the LMCO family was examined using TBtools for the investigation of evolutionary history [44].

### 4.4. Study of Gene and Protein Motifs

Conserved motifs in the full-length LMCO protein sequences were identified using MEME tools (http://meme-suite.org/tools/meme) (accessed on 29 December 2022) [45,46]. The Gene Structure Display Server (GSDS 2.0) (http://gsds.cbi.pku.edu.cn/) (accessed on 30 December 2022) was utilized to study the structure of the LMCO gene following the standard procedure of Hu et al. [47].

### 4.5. LMCO Gene–Protein Structure Analysis

The online secondary structure prediction tool SOPMA (https://npsa-prabi.ibcp.fr/cgi-in/secpredsopma.pl) (accessed on 3 January 2022) was used to determine the structural makeup of the 18 Br-Lac genes. Similarly, the protein structural makeup of the 18 LMCO genes was also determined through an online tool (https://swissmodel.expasy.org/) (accessed on 7 January 2022) [48].

### 4.6. Protein Physicochemical Properties of LMCO

The NCBI database (https://www.ncbi.nlm.nih.gov/) (accessed on 7 January 2022) was used to determine amino acid (bp), CDS (bp), and location on the chromosome of the LMCO gene to clarify its physicochemical characteristics. Isoelectric points (pI), molecular weight (MW), grand average of hydropathicity (GRAVY), and the formula of the individual LMCO gene in *B. rapa* were assessed through the ProtParam tool on the ExPASy server (http://web.expasy.org/protparam) (accessed on 29 April 2023). We also used the WoLF PSORT service for plant protein location (https://wolfpsort.hgc.jp/) (accessed on 29 April 2023) to assess LMCO protein in *Brassica* [49].

### 4.7. Calculating Ka and Ks

The substitution rates of Ka and Ks of the syntenic gene pairs were measured with Nei–Gojobori using TBtools software. We used the KaKs calculator 2.0 with the Nei–Gojobori method, as practiced by several workers [48,49,50].

### 4.8. Plant Materials and Stress Treatments

*B. rapa* seeds were soaked in water (2 days) for the process of germination. The soaked seeds were then cultured in 10 cm plastic pots. The germinated seedlings were then allowed to grow in a chamber room under light and dark (16/8) at 25 °C until the four-leaf stage. Different abiotic stresses (i.e., drought, heat, and salinity) were applied to the growing seedlings [51]. For drought conditions, the germinated seedlings were treated with 20% polyethylene glycol-6000 (PEG-6000), while 200 mM NaCl was used to initiate salinity stress. For heat stress, the germinated seedlings were exposed to a high temperature of 40 °C. To calculate the seedlings’ response to drought stress, we collected leaf samples after 3, 6, 12, 24, 48, and 72 h [52]. In a similar pattern, young leaves were collected after 3, 6, 12, 24, and 48 h to document resistance to salinity stress. After collection, the leaf samples were kept in liquid nitrogen and stored at −80 °C for RNA isolation.

### 4.9. RNA Extraction and Real-Time qRT-PCR

A Plant Total RNA Isolation Kit (FOREGENE, China) was used for total RNA extraction from differently treated samples (i.e., drought, salinity, or heat). A RevertAid First Strand cDNA Synthesis Kit (Thermo Fisher Scientific, (Waltham, Massachusetts, USA) was used to synthesize purified cDNA with 1 μg of total RNA and oligo primers [53]. Light Cycler^®^ 480 II (Mannheim, Roche, Germany) and SYBR Green I Master Mix (Roche, Germany) were used to perform qPCR. The gene-specific primers were designed using the NCBI online software primer designing tool (https://www.ncbi.nlm.nih.gov/tools/primer-blast/) (accessed on 15 January 2023) (Appendix A). Each qPCR reaction was conducted as follows: 0.2 μL of cDNA, 5 μL of 0.5 μM gene-specific primer pre-mixture, 10 μL of 2 × SYBR Green Master Mix, and 4.8 μL of water. Actin7 was used as the internal standard to normalize the expression levels for target genes. A melting curve was used to evaluate the specificity of amplification. All experiments had three biological replicates and technical replicates. The 2−ΔCT method was used for data calculation. Graph Pad Prism 9 was used to analyze and graph the expression data.

## 5. Conclusions

In conclusion, 18 LMCO genes were found in *B. rapa* during this investigation. They were categorized into seven groups based on phylogenetic analysis and similarities in amino acid sequences. The basic gene parameters, such as amino acid and CDS length, molecular weight (MW/kDa), isoelectric point (PI), and GRAVY, as well as the fact that the majority of Br-Lac genes were located in chloroplasts, were unearthed by the physicochemical properties of Br-Lac genes. The divergence study further illuminated the Br-Lac genes’ evolutionary history, which revealed that the time of divergence ranged from 12.365 to 39.250 MYA. These findings point to a shared biological function of Br-Lac genes in response to enzymatic activity and offer helpful hints for further research into the diversity and function of Br-Lac genes. Expression analyses of the Br-Lac gene family were performed in this study for abiotic stress. The findings suggest that Br-lac genes resist heat, drought, and salinity stress.

## Figures and Tables

**Figure 1 plants-12-01904-f001:**
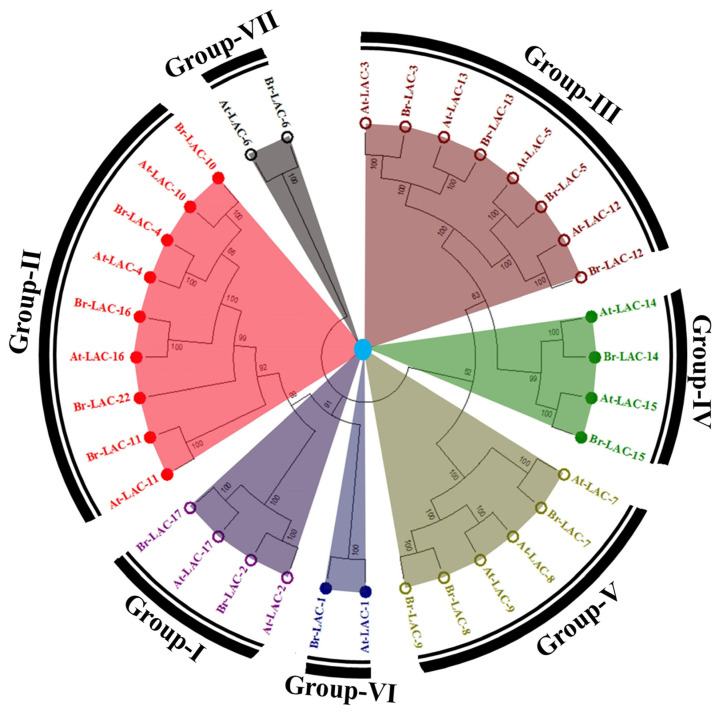
Identification and phylogenetic analysis of LMCO gene family members in *B. rapa* L. (18) against *A. thaliana* (17) using maximum likelihood methods in MEGA7 for 35 LMCO protein sequences from *A. thaliana* and *B. rapa* L. The bootstrap consensus tree was generated using the Poisson model with 1000 bootstraps.

**Figure 2 plants-12-01904-f002:**
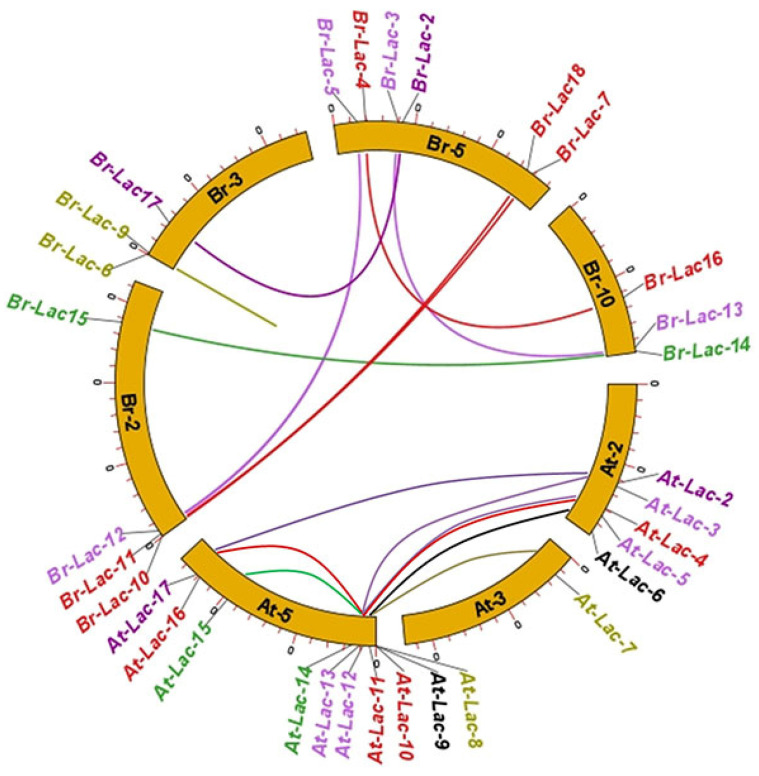
Localization and synteny of LMCO genes in *Arabidopsis* and *B. rapa* L. Gene pairs with syntenic relationships are joined by colored lines.

**Figure 3 plants-12-01904-f003:**
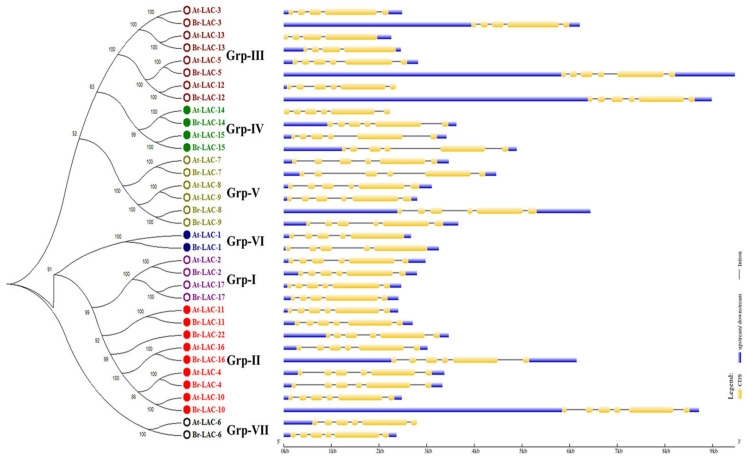
Phylogenetic analysis and schematic diagram for intron/exon structures of LMCO genes from *B. rapa* L. and *A. thaliana*. The phylogenetic tree was constructed from a complete alignment of 35 LMCO proteins using the maximum likelihood method with bootstrapping analysis (1000 iterations). The orange, blue, and black colors in the gene structure diagram represent exons, introns, and UTR, respectively. Gene models are drawn to scale, as indicated at the bottom.

**Figure 4 plants-12-01904-f004:**
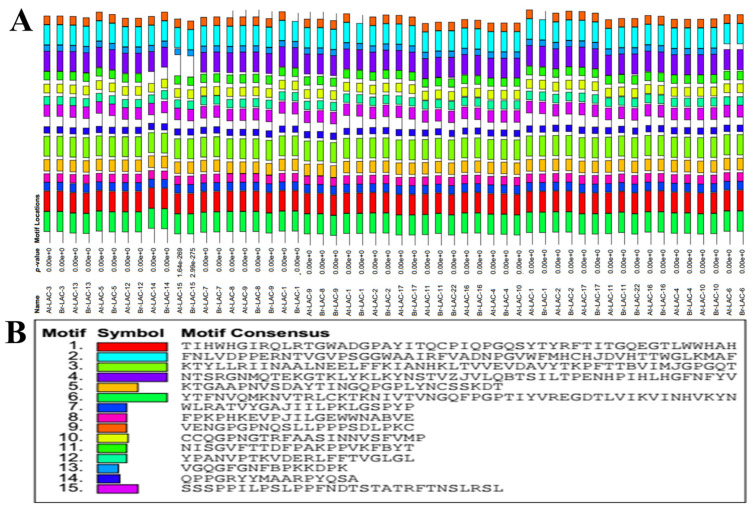
Motifs and sequence logos identified using MEME tools in the LMCO gene family of *B. rapa* L. and *A. thaliana.*(**A**) Identified motifs; (**B**) Location and combined *p*-values of the total 15 motifs in the LMCO genes of *B. rapa* L. and *A. thaliana* using the MEME tool. The scale indicates the lengths and motifs of the proteins.

**Figure 5 plants-12-01904-f005:**
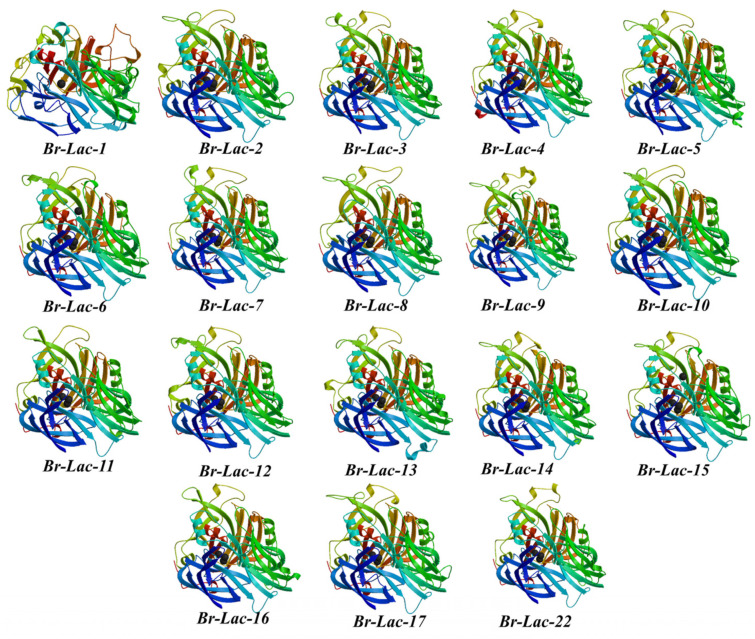
Illustration of the predicted tertiary structures of the 18 LMCO proteins of *B. rapa* L. The protein structures all have the same domain color schemes, revealing the degree of homology. The structures reveal a high degree of structural homology in most gene members.

**Figure 6 plants-12-01904-f006:**
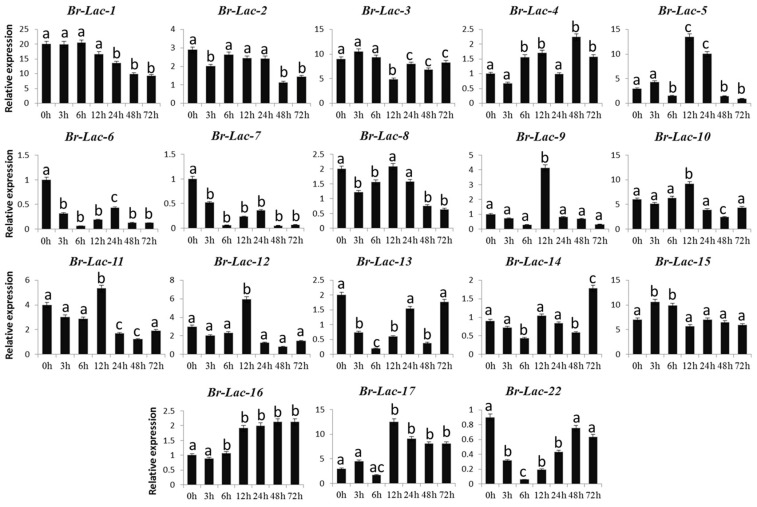
Expression analysis of Br-Lac (LMCO) gene family members under polyethylene glycol (PEG)-induced drought stress in *B. rapa*. Different letters indicate significant differences, as determined by ANOVA followed by LSD and Tukey’s multiple comparison test (*p* < 0.05). Data are presented as mean ± SEM.

**Figure 7 plants-12-01904-f007:**
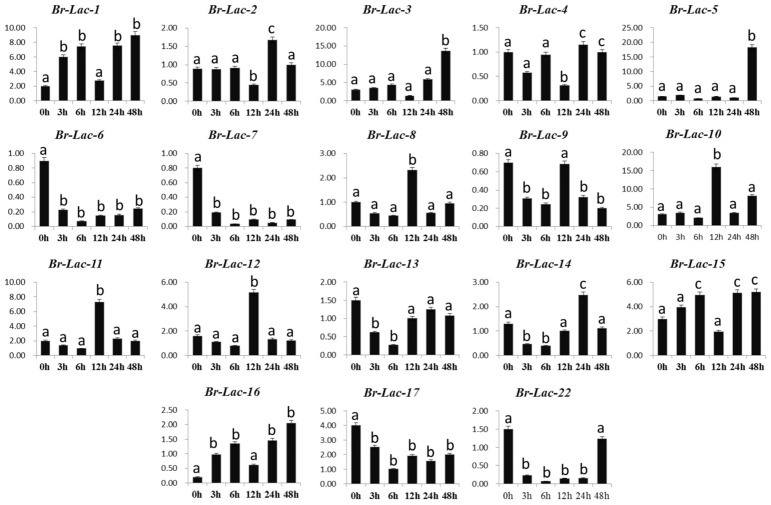
Expression analysis of Br-Lac (LMCO) gene family members under salinity (salt) stress in *B. rapa*. Different letters indicate significant differences, as determined by ANOVA followed by LSD and Tukey’s multiple comparison test (*p* < 0.05). Data are presented as mean ± SEM.

**Figure 8 plants-12-01904-f008:**
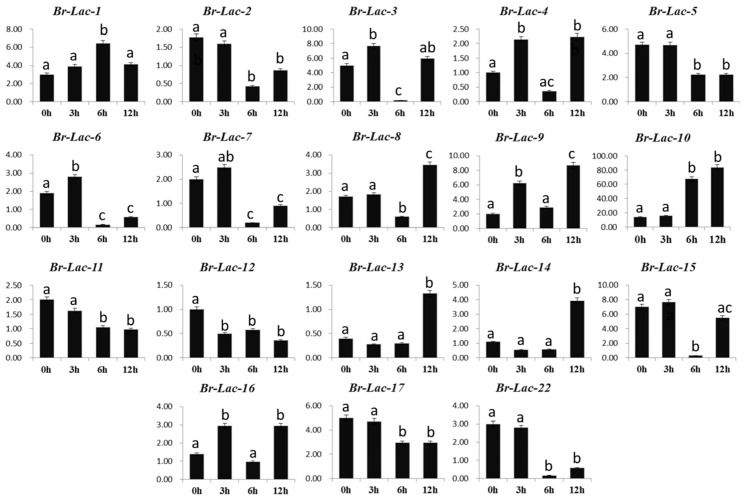
Expression analysis of Br-Lac (LMCO) gene family members under heat (high temperature) stress in *B. rapa*. Different letters indicate significant differences, as determined by ANOVA followed by LSD and Tukey’s multiple comparison test (*p* < 0.05). Data are presented as mean ± SEM.

**Table 1 plants-12-01904-t001:** Characteristics of LMCO genes from *B. rapa* L.

Gene Name	Accession ID	Chromosome	CDS Length (bp)	Protein Length (aa)	Molecular Weight (kDa)	Isoelectric Point	GRAVY	Subcellular Localization
*Br-LAC-1*	XP_009110363.1	A08	2023	579	65.13	9.12	−0.258	Chloroplast
*Br-LAC-2*	XP_009144396.1	A05	2284	578	64.23	9.47	−0.091	Vacuole
*Br-LAC-3*	XP_009144319.1	A05	5873	570	63.99	9.46	−0.277	Chloroplast
*Br-LAC-4*	XP_033148764.1	A05	2076	558	61.57	9.36	−0.089	Extracellular
*Br-LAC-5*	XP_009143220.1	A05	8357	574	63.75	8.99	−0.158	Vacuole
*Br-LAC-6*	XP_009142428.1	A04	2013	569	63.67	7.96	−0.178	Chloroplast
*Br-LAC-7*	XP_009146941.1	A05	2274	568	62.66	8.94	0.015	Chloroplast
*Br-LAC-8*	XP_033142360.1	A03	5279	586	65.25	7.29	−0.056	Cytoplasmic
*Br-LAC-9*	XP_009132200.3	A03	2538	582	64.81	6.07	−0.05	Chloroplast
*Br-LAC-10*	XP_009125430.1	A02	7379	559	61.23	9.49	−0.034	Chloroplast
*Br-LAC-11*	XP_009125488.1	A02	2126	560	62.05	8.63	−0.069	Extracellular
*Br-LAC-12*	XP_009125599.1	A02	8316	565	62.65	9.52	−0.165	Chloroplast
*Br-LAC-13*	XP_009122370.1	A10	2223	566	62.91	6.52	−0.149	Endoplasmic reticulum
*Br-LAC-14*	XP_009122510.1	A10	2830	580	65.76	9.73	−0.301	Chloroplast
*Br-LAC-15*	XP_009129702.1	A02	3080	559	63.49	9.15	−0.208	Vacuole
*Br-LAC-16*	XP_009120399.1	A10	4954	565	62.52	9.15	−0.052	Chloroplast
*Br-LAC-17*	XP_009131958.1	A03	2101	573	63.67	9.18	−0.15	Chloroplast
*Br-LAC-22*	XP_009146739.1	A05	2760	560	61.82	8.47	−0.079	Chloroplast

**Table 2 plants-12-01904-t002:** Non-synonymous (Ka) and synonymous (Ks) substitution rate and divergence time of LMCO genes.

Node Number	Gene Pairs	Ka	Ks	Ka/Ks Ratio	Time (MYA)
1	*(Br-Lac-1)-(Br-Lac-6)*	0.224	0.533	0.420	17.770
2	*(Br-Lac-17)-(Br-Lac-2)*	0.104	0.550	0.188	18.338
4	*(Br-Lac-16)-(Br-Lac-4)*	0.078	0.412	0.190	13.728
5	*(Br-Lac-22)-(Br-Lac-11)*	0.129	0.418	0.309	13.932
8	*(Br-Lac-7)-(Br-Lac-10)*	0.182	0.446	0.408	14.882
9	*(Br-Lac-9)-(Br-Lac-8)*	0.025	0.118	0.213	39.250
10	*(Br-Lac-15)-(Br-Lac-14)*	0.167	0.404	0.415	13.455
13	*(Br-Lac-12)-(Br-Lac-5)*	0.065	0.371	0.176	12.365
14	*(Br-Lac-13)-(Br-Lac-3)*	0.076	0.396	0.191	13.183

Ks = synonymous substitution rate, Ka = non-synonymous substitution rate, MYA = million years ago. Using a divergence rate of 1.5 × 10^−9^ mutations per synonymous site per year.

**Table 3 plants-12-01904-t003:** Secondary structure of 18 Br-Lac proteins.

Genes	H (%)	T (%)	E (%)	RC (%)
*Br-Lac-1*	14.88	11.59	33.39	40.14
*Br-Lac-2*	14.71	10.21	31.14	43.94
*Br-Lac-3*	16.84	10.88	28.77	43.51
*Br-Lac-4*	16.49	10.39	32.62	40.5
*Br-Lac-5*	18.82	9.41	30.84	40.94
*Br-Lac-6*	18.28	8.79	30.4	42.53
*Br-Lac-7*	18.84	11.09	28.87	41.2
*Br-Lac-8*	18.09	9.04	30.72	42.15
*Br-Lac-9*	12.2	9.11	32.99	45.7
*Br-Lac-10*	21.11	11.45	30.77	36.67
*Br-Lac-11*	14.11	10.71	33.75	41.43
*Br-Lac-12*	15.22	8.67	31.5	44.6
*Br-Lac-13*	15.72	9.19	31.8	43.29
*Br-Lac-14*	18.1	11.03	31.55	39.31
*Br-Lac-15*	14.49	9.3	30.77	45.44
*Br-Lac-16*	16.46	10.27	30.09	43.19
*Br-Lac-17*	15.71	8.55	30.54	45.2
*Br-Lac-22*	20.89	9.82	29.46	39.82

## Data Availability

The datasets generated and analyzed during the current study are available from the corresponding author upon reasonable request.

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
