# Peer review of "A Genome-Wide Identification and Expression Pattern of LMCO Gene Family from Turnip (Brassica rapa L.) under Various Abiotic Stresses"

_plants, 2023, doi:10.3390/plants12091904_

Round 1
Reviewer 1 Report
The authors report a molecular analysis of genes related to oxidase enzymes in Brassica rapa under water stress.
The authors first performed gene sequence analysis of Brassica rapa and Arabidopsis thaliana, then applied water stress to B. rapa seedlings and examined LMCO gene expression. These enzymes are essential for the synthesis of lignin and other plant secondary metabolites. In particular, laccase-like multi-copper oxidase (LMCO) is a type of enzyme that catalyses the oxidation of various substrates using copper ions. This enzyme is called "laccase-like" because it has similarities in structure and function to the well-known enzyme laccase, which is also a multi-copper oxidase. LMCOs enzymes play important roles in various biological processes such as lignin degradation, melanin biosynthesis and wound healing. LMCOs have been extensively studied for their potential applications in various fields such as bioremediation, biocatalysis and bioenergy production. They have shown promising results in wastewater treatment and in the production of high-value chemicals, such as vanillin and ferulic acid, from lignocellulosic biomass.
The manuscript is well written, it is clear and a lot of experimental data has been collected. The study of these enzymes is essential to understand the response of plants to drought stress and the authors reported very important data for agronomic studies of plants.
In my opinion, the manuscript has only typographical errors. In addition, the names of plant species should always be written in italics.
After these minor changes, the manuscript can be accepted.
Author Response
Point 1: The manuscript is well written, it is clear and a lot of experimental data has been collected. The study of these enzymes is essential to understand the response of plants to drought stress and the authors reported very important data for agronomic studies of plants.
Response: We are thankful to the anonymous reviewer for keenly studying our manuscript and encouraging us for scientific research in this field.
Point 2: In my opinion, the manuscript has only typographical errors. In addition, the names of plant species should always be written in italics.
Response: The manuscript is now revised and all the suggested changes are incorporated successfully.
Point 3: In addition, the names of plant species should always be written in italics.
Response: Plant species names are cross checked and corrected as per binomial nomenclature roles.
Point 4: After these minor changes, the manuscript can be accepted.
Response: The authors are tankful to the anonymous reviewer for showing interest in the manuscript.

Reviewer 2 Report
In the manuscript “A genome-wide identification and expression pattern of LMCO gene family from Turnip (Brassica rapa L.) under various abiotic stresses" Khan et al. reported the thoroughly computational analysis of 18 LMCO genes in turnip, and tested the transcriptions upon the abiotic stresses. The experiments were not well conducted, controls and more shreds of evidence are needed for most of the conclusions, The authors should revise the language and logic of manuscript to improve readability. As explained below, the concerns regarding the experimental design need to be addressed in order for the main conclusions to be well-founded.
1. Line 22, “cur-rent” should be “current”.
2. Line 30, “responds” should be “respond”.
3. Line 36 “Multi-copper oxidase (MCOs) is a group of enzymes…” should be “Multi-copper oxidases (MCOs) are a group of enzymes…”
4. Line 41 “…lignin and flavonoids and anthocyanin degradation.” Should be “…lignin, flavonoids and anthocyanin degradation.”
5. Line 77-79, “This study's findings will help lay the groundwork for future research on its biological function. ” “help lay” should be “help laying”.
6. Line 102, in Figure 2 title LMCO should the italic.
7. Figure 2 please redo figure 2 to segmentally duplicated LMCO genes and the connecting lines. You can also include Arabidopsis LMCOs to show Arabidopsis – Brassica rapa synteny. Why the Br-Lac6 is not connected to any gene?
8. Please correct in the form of genes in italic throughout the whole draft.
9. Line 126 what is the “LAMCO”?
10. What are the 12th color bars stand for in At-LAC-15 and Br-LAC-15?
11. Please reorganize the consensus motifs in Fig.4 B, it is not easy for the readers to interpret in the current version. Please use more symbol styles instead of using only colors.
12. Line 186, how old are the healthy seedlings? What part of the plants were used to perform the qRT-PCR?
13. Line 186, Brassica rapa should be italic.
14. Figure 6, please add bars on Y axis as Br-Lac-1.
15. Figure 6, please what is the normalized control? What is experimental treatment control?
16. Figure 6 please also add the statistical analysis.
17. Please separate three abiotic treatments for the qRT-PCR analysis.
18. In Figure 6, please put more controls to exclude the possibility of circadian rhythm effects on these genes.
19. Please use the same style eighter B. rapa or Brassica rapa throughout the manuscript.
Author Response
Point 1: In the manuscript “A genome-wide identification and expression pattern of LMCO gene family from Turnip (Brassica rapa L.) under various abiotic stresses" Khan et al. reported the thoroughly computational analysis of 18 LMCO genes in turnip, and tested the transcriptions upon the abiotic stresses. The experiments were not well conducted, controls and more shreds of evidence are needed for most of the conclusions, The authors should revise the language and logic of manuscript to improve readability. As explained below, the concerns regarding the experimental design need to be addressed in order for the main conclusions to be well-founded.
Response: We agree with the reviewer comments and the manuscript is now revised according to the reviewer suggestions, which improved its scientific quality.
Point 2: Line 22, “cur-rent” should be “current”.
Response: Corrected as suggested by the reviewer.
Point 3: Line 30, “responds” should be “respond”.
Response: Corrected in the revised version of the manuscript.
Point 4: Line 36 “Multi-copper oxidase (MCOs) is a group of enzymes…” should be “Multi-copper oxidases (MCOs) are a group of enzymes e 22, “cur-rent” should be “current”.
Response: Corrected as suggested by the reviewer.
Point 5: Line 41 “…lignin and flavonoids and anthocyanin degradation.” Should be “…lignin, flavonoids and anthocyanin degradation.”
Response: Corrected as suggested by the reviewer.
Point 6: Line 77-79, “This study's findings will help lay the groundwork for future research on its biological function” “help lay” should be “help laying”.
Response: Corrected as suggested by the reviewer.
Point 7: Line 102, in Figure 2 title LMCO should the italic.
Response: Corrected as suggested by the reviewer.
Point 8: Figure 2 please redo figure 2 to segmentally duplicated LMCO genes and the connecting lines. You can also include Arabidopsis LMCOs to show Arabidopsis – Brassica rapa synteny. Why the Br-Lac6 is not connected to any gene?
Response: Figure 2 is modified as per reviewer suggestion. The Arabidopsis LMCOs genes were also included and re-performed. Now “Figure 2. “Localization and synteny of LMCO genes in Arabidopsis and B. rapa L. Gene pairs with syntenic relationships are joined by the colored lines”. The Br-Lac6 gene is connected with Br-Lac9; both genes are located on the same chromosome.
Point 9: Please correct in the form of genes in italic throughout the whole draft.
Response: Whole manuscript is revised and changes were as suggested by the reviewer.
Point 10: Line 126 what is the “LAMCO”?
Response: It was typographical error, now corrected in the revised version of the manuscript.
Point 11: What are the 12th color bars stand for in At-LAC-15 and Br-LAC-15?
Response: Figure 4. 12th color bars stand for motif number 4. Motif 4 is missing in At-LAC15 and Br-LAC-15 that’s why the original color of motifs 4 becomes less (Shadow color). 12th color bars is the missing color of motif 4. Now the figure number 4 analysis is re-performed and (Shadow color or missing color) were not mentioned in the new figure (Figure 4).
Point 12: Please reorganize the consensus motifs in Fig.4 B, it is not easy for the readers to interpret in the current version. Please use more symbol styles instead of using only colors.
Response: Figure 4 is auto-generated data from https://meme-suite.org/meme/tools/meme, the colors are usually used for motifs study. Motifs data were also given in the supplementary materials for detail study. Which make all the analysis easy to read. However, we have revise the figure 4 with more good resolution to make it better.
Point 13: Line 186, how old are the healthy seedlings? What part of the plants were used to perform the qRT-PCR?
Response: The germinated seedling was allowed to grow in a chamber room (16 hours light and 8 hours dark per day at 25 °C) till four leaf stage. After 18 to 20 days of four leaf stage, different abiotic stresses, i.e., drought, heat and salinity, were applied to the B. rapa seedlings. With different time intervals (mentioned in material and methods line 318-328) the leaves for collected for qRT-PCR (Line number 334-335).
Point 14: Line 186, Brassica rapa should be italic.
Response: Corrected as suggested by the reviewer. This action is performed throughout the draft.
Point 15: Figure 6, please add bars on Y axis as Br-Lac-1.
Response: As per previous comments: Figure 6 split into 3 figure as per abiotic treatments, i.e., fig. 6, fig. 7, and fig. 8. Bars were added to Y axis in all graphs accordingly.
Point 16: Figure 6, please what is the normalized control? What is experimental treatment control?
Response: Actin7 was used as the internal standard to normalize the expression level. Actin7 is used in qRT-PCR as a normalized control in the experiment. 0h (zero hour/ 0-h) samples were collected as control before treatment. The 0h (Zero hour) samples were used as an experimental treatment control.
Point 17: Figure 6 please also add the statistical analysis.
Response: As per previous comments: Figure 6 split into 3 figure as per abiotic treatments, i.e., fig. 6, fig. 7, and fig. 8. The statistical analysis were performed and mentioned in fig 6-8.
Point 18: Please separate three abiotic treatments for the qRT-PCR analysis.
Response: As per reviewer comments, the different abiotic treatments for qRT-PCR were separated from Figure 6 to new figure number (Figure 6, Figure 7 and Figure 8).
Point 19: In Figure 6, please put more controls to exclude the possibility of circadian rhythm effects on these genes.
Response: Figure 6, the 0h (zero hour) samples used as control. For qRT-PCR the Actin7 is used as internal control.
Point 20: Please use the same style eighter B. rapa or Brassica rapa throughout the manuscript.
Response: Revised and corrected as per international scientific roles.

Reviewer 3 Report
The manuscript under review is devoted to the identification of possible patterns of expression of the LMCO genes family in response to the impact of abiotic stress factors - drought, salinity, and exposure to high temperatures in Brassica rapa plants. The LMCO family of genes encodes a wide range of laccase-like multicopper oxidases, which attract the attention of researchers due to their ability to oxidize certain substrates. The primary bioinformatic analysis of this family of genes in Brassica rapa seems to be very relevant, since some members of this family may be involved in plant responses to abiotic stresses, in particular, drought, high temperatures, and salinity. It is this problem that is addressed in this manuscript. The authors of this manuscript carried out an evolutionary analysis of conserved domains and motifs, analyzed the structure and organization of LMCO family genes in rapeseed. A total of 18 genes of this family were identified, which were clustered into 7 groups. The genes of the LMCO family identified by the authors were studied in terms of changes in their expression in response to the impact of such stress factors as drought, salinity, and heat. Based on the analysis of qRT-PCR data, the data of both increase and decrease in the expression level for a number of genes of this family are presented. When conducting rapeseed responses to stressful influences, the authors used rapeseed plants at the stage of 4 formed leaves, however, in the description of plant material, the authors did not indicate the species and varietal affiliation of the analyzed material. Which variety or natural sample was used in the analysis, its origin? In general, the authors used modern approaches to bioinformatics analysis with the involvement of modern databases and software packages. The results of the analysis are presented in the form of figures, there are no comments on their design. The conducted bioinformatic analysis will serve in the future to identify the biological functions of the genes of the family under study. The manuscript can be recommended for publication, taking into account the comments made.
Author Response
Point 1: The LMCO family of genes encodes a wide range of laccase-like multicopper oxidases, which attract the attention of researchers due to their ability to oxidize certain substrates. The primary bioinformatic analysis of this family of genes in Brassica rapa seems to be very relevant, since some members of this family may be involved in plant responses to abiotic stresses, in particular, drought, high temperatures, and salinity. It is this problem that is addressed in this manuscript
Response: The authors are thankful to the anonymous reviewer for keenly studying the manuscript.
Point 2: When conducting rapeseed responses to stressful influences, the authors used rapeseed plants at the stage of 4 formed leaves, however, in the description of plant material, the authors did not indicate the species and varietal affiliation of the analyzed material. Which variety or natural sample was used in the analysis, its origin?
Response: In this research work the Brassica rapa L. Turnip (Brassica rapa var. rapa L.) were selected, the seed were collected from local market in Pakistan.
Point 3: The manuscript can be recommended for publication, taking into account the comments made.
Response: We are thankful to the anonymous reviewer for encouraging us for scientific research in this field.

Round 2
Reviewer 2 Report
I appreciate the authors worked hard to address some of my concerns. However, the overall scientific question and hypothesis testing to uncover the mechanism behind it are still not sound. The provided data do not fully support the significant conclusion. The major figures are in silico, and the wet experiment, the qRT-PCR is not properly analyzed, and controls are needed to claim the conclusions. For example, when performing PEG treatment, a mock control is required in order to exclude the circadian rhythm effect on the Br-Lacs expression, also, the expression level of the 0h sample should be normalized to 1, etc. Besides, tons of RNA-seq data are available to extract the gene of interest responsive to a variety of stimuli. In all, this draft does not provide much new and convincing information with its figures to the scientific community.